# Practical and stereoselective electrocatalytic 1,2-diamination of alkenes

Chen-Yan Cai[1], Xiao-Min Shu[1] & Hai-Chao Xu [1]*

The 1,2-diamine motif is widely present in natural products, pharmaceutical compounds, and catalysts used in asymmetric synthesis. The simultaneous introduction of two amino groups across an alkene feedstock is an appealing yet challenging approach for the synthesis of 1,2-diamines, primarily due to the inhibitory effect of the diamine products to transition metal catalysts and the difficulty in controlling reaction diastereoselectivity and regioselectivity. Herein we report a scalable electrocatalytic 1,2-diamination reaction that can be used to convert stable, easily available aryl alkenes and sulfamides to 1,2-diamines with excellent diastereoselectivity. Monosubstituted sulfamides react in a regioselective manner to afford 1,2-diamines bearing different substituents on the two amino groups. The combination of an organic redox catalyst and electricity not only obviates the use of any transition metal catalyst and oxidizing reagent, but also ensures broad reaction compatibility with a variety of electronically and sterically diverse substrates.

---

[1] State Key Laboratory of Physical Chemistry of Solid Surfaces, Innovation Center of Chemistry for Energy Materials, Key Laboratory of Chemical Biology of Fujian Province, and College of Chemistry and Chemical Engineering, Xiamen University, Xiamen 361005, People's Republic of China. *email: haichao.xu@xmu.edu.cn

1,2-Diamine is a prevalent structural motif in natural products, pharmaceutical compounds, and molecular catalysts[1]. Alkene 1,2-diamination and 1,2-diazidation reactions are among the most straightforward and attractive strategies for 1,2-diamine synthesis, especially considering the easy accessibility and handling of alkene substrates[2]. Significant progress has been achieved over the past decades in alkene 1,2-diamination and 1,2-diazidation reactions, mainly through transition metal catalysis (Fig. 1a, b)[3–18]. Unfortunately, these methods are not without drawbacks. First, the use of stoichiometric amounts of transition metal reagents (e.g., osmium or cobalt)[3,7], chemical oxidants (e.g., iodine (III) reagents or organic peroxides)[5,6,10,11], or azide reagents[8–14] raises cost, environmental, and safety issues, especially for large-scale applications[19,20]. Second, they are often limited in substrate scope, sometimes requiring special amination reagents (e.g., diaziridinone and its analogs[4,15,16], or azido-iodine compounds[9]). Other challenges that need to be addressed include unsatisfactory diastereoselectivity for internal alkenes and

**Fig. 1** Synthesis of 1,2-diamines. **a, b** Representative examples of established 1,2-diamine synthesis via vicinal difunctionalization of alkenes. **c** Proposed electrochemical 1,2-diamination of alkenes with sulfamides via dehydrogenative annulation and removal of the sulfonyl group. Boc, *tert*-butyloxycarbonyl; Ms, methanesulfonyl; TMS, trimethylsilyl

poor differentiation of the two amino groups in the diamine products.

Organic electrochemistry, which drives redox processes with electric current, is increasingly considered as a highly sustainable and efficient synthetic method[21–34]. One key advantage of using electrochemical methods is that the reaction efficiency and selectivity can often be boosted by manipulating the electric current or potential, allowing one to achieve transformations that are otherwise synthetically inaccessible. In this context, Yoshida[35] reported isolated examples of alkene diamination through intramolecular trapping of alkene radical cations. Shäfer[36] reported an early example of electrochemical 1,2-diazidation of simple alkenes with NaN$_3$ in acetic acid. Lin and co-workers recently developed a NaN$_3$-based electrocatalytic olefin 1,2-diazidiation reaction that showed an exceptional substrate scope and broad functional group compatibility (Fig. 1b, bottom)[12–14].

Building on our experience with electrochemical alkene difunctionalization[37,38], herein we report a diastereoselective electrocatalytic 1,2-diamination reaction of di- and tri-substituted alkenes using easily available and stable sulfamides as amino donors. A wide variety of 1,2-diamines, where the two amino groups are functionalized with different substituents, can be prepared via regio- and diastereoselective diamination using monosubstituted sulfamides. The electrosynthetic method employs an organic redox catalyst and proceeds through H$_2$ evolution, while obviating the need for transition metal catalysts and external chemical oxidants.

## Results

**Design plan.** Inspired by our previously work on electrochemical alkene dioxygenation reactions[38], we envisioned the trapping of an electrocatalytically generated alkene radical cation **II•+** with a sulfamide **III** to generate a carbon radical **IV** (Fig. 2a)[39–43]. Single-electron transfer oxidation of **IV** by **I•+** would produce a carbocation **V**, which could then undergo cyclization to afford the cyclic sulfamide **VI**. Cyclization of **V** has a key role in governing the stereoselectivity of the 1,2-diamination, in which the alkene-originated substituents R$^1$ and R$^2$ are positioned on opposite sides of the nascent five-membered ring to minimize steric repulsion.

The electrons that the alkene loses to the anode would eventually combine with the protons at the cathode to form H$_2$, thereby obviating the need for external electron and proton acceptors. The controlled formation of alkene radical cations at low concentrations is essential to overcome their strong propensity toward self-dimerization or reaction with the alkene precursors, especially on electrode surface[44–46]. This could be accomplished by conducting electrolysis indirectly in the presence of a redox catalyst rather than direct electrolysis. Measuring catalytic current through cyclic voltammetry[33,47], with tris(2,4-dibromophenyl) amine (**1**, $E_{p/2} = 1.48$ V vs saturated calomel electrode (SCE)) as the redox catalyst, confirmed the facile electrocatalytic oxidation of the alkenes **2** ($E_{p/2} = 1.66$ V vs SCE) and **3** ($E_{p/2} = 1.80$ V vs SCE) that bears an electron-withdrawing ester group (Fig. 2b, c).

**Reaction optimization.** The 1,2-diamination of aryl alkene **2** with sulfamide **4** was chosen as a model reaction for optimizing the electrochemical conditions. The electrolysis was conducted at RT and at a constant current in a three-necked round-bottomed flask equipped with a reticulated vitreous carbon (RVC) anode and a platinum plate cathode. The optimal reaction system consisted of triarylamine **1** (10 mol %) as redox catalyst, $^i$PrCO$_2$H (2 equiv) and BF$_3$•Et$_2$O (0.5 equiv) as additives, Et$_4$NPF$_6$ as supporting electrolyte to increase conductivity, and MeCN/CH$_2$Cl$_2$ (1:2) as solvent. Under these conditions, the diamination product **5** was obtained in 72% yield with excellent diastereoselectivty ( > 20:1 dr) even though a starting mixture of Z/E isomers of **2** was used (in a ratio of 5.6:1) (Table 1, entry 1). Independent reaction using a pure E- or Z-isomer of **2** afforded the same trans diastereomer **5** in 65% and 69% yield, respectively. Similar results could also be obtained when the reaction was performed in ElectraSyn 2.0, a commercial apparatus (Table 1, entry 2). The use of MeCN as solvent instead of MeCN/CH$_2$Cl$_2$ resulted in a lower yield of 50% (Table 1, entry 3). Other triarylamine derivatives such as **6** ($E_{p/2} = 1.06$ V vs SCE), **7** ($E_{p/2} = 1.26$ V vs SCE), and **8** ($E_{p/2} = 1.33$ V vs SCE) were found to be less effective in promoting the formation of **5** probably because of their lower oxidation potentials (Table 1, entries 4–6). Control experiments showed that the triarylamine catalyst (Table 1, entry

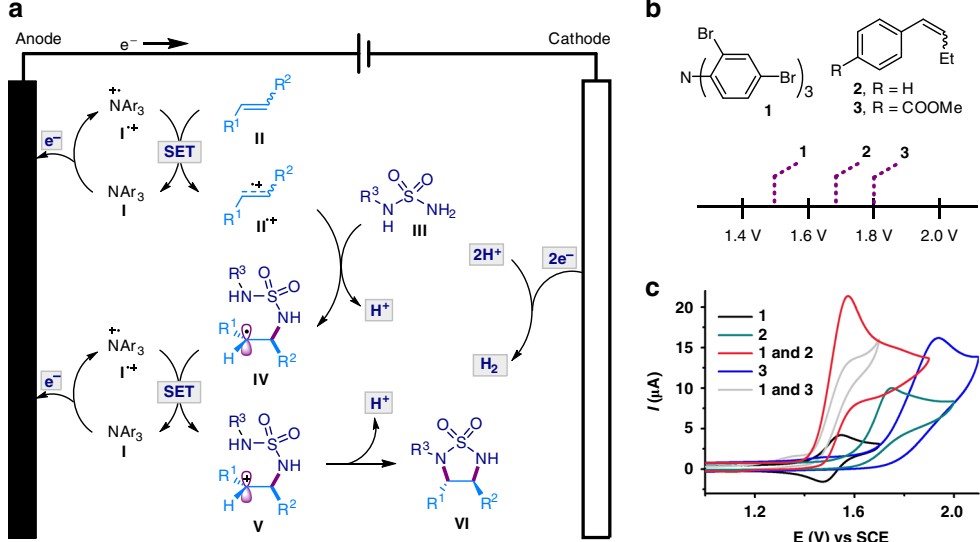

**Fig. 2** Proposed reaction design. **a** The proposed reaction mechanism. The process combines anodic oxidation and cathodic proton reduction to achieve the alkene 1,2-diamination via H$_2$ evolution. The electrocatalytic activation of the alkene through single-electron transfer (SET) oxidation generates the alkene radical cation **II•+**, which is trapped by the sulfamide **III** to give radical **IV**. Further SET oxidation and diastereoselective cyclization of **V** afford diamination product **VI**. The cathode reduces protons to generate H$_2$. **b** Oxidation potentials [$E_{p/2}$ vs Saturated calomel electrode (SCE)] of triarylamine **1** and alkenes **2** and **3**. **c** Cyclic voltammetry. The studies show that triarylamine **1** can catalyze the oxidation of aryl alkenes **2** and **3**

**Table 1 Optimization of reaction conditions**[a]

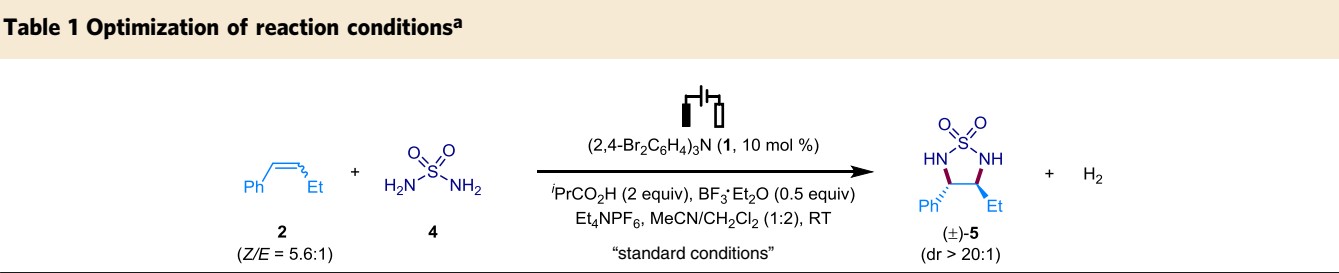

| Entry | Deviation from standard conditions | Yield of 5 (%)[b] |
|---|---|---|
| 1 | None | 72[c] |
| 2 | Reaction conducted using ElectraSyn 2.0 | 74 |
| 3 | MeCN as solvent | 50 |
| 4 | (4-BrC$_6$H$_4$)$_3$N (**6**) as the catalyst | 16 |
| 5 | (4-MeO$_2$CC$_6$H$_4$)$_3$N (**7**) as the catalyst | 51 |
| 6 | (2,4-Br$_2$-C$_6$H$_3$)$_2$N(4-Br-C$_6$H$_4$) (**8**) as the catalyst | 52 |
| 7 | No **1** | 23 |
| 8 | No $^i$PrCO$_2$H | 58 |
| 9 | No BF$_3$•Et$_2$O | 46 |
| 10 | AcOH instead of $^i$PrCO$_2$H | 65 |
| 11 | CF$_3$CO$_2$H instead of $^i$PrCO$_2$H | 67 |
| 12 | No $^i$PrCO$_2$H and BF$_3$•Et$_2$O | 20 |
| 13 | HBF$_4$ (0.5 equiv) instead of $^i$PrCO$_2$H and BF$_3$•Et$_2$O | 66 |

[a]Reaction conditions: RVC (100 PPI, 1 cm × 1 cm × 1.2 cm), Pt plate cathode (1 cm × 1 cm), **2** (0.2 mmol), **4** (0.4 mmol), MeCN (2 mL), CH$_2$Cl$_2$ (4 mL), Et$_4$NPF$_6$ (0.2 mmol), 12.5 mA ($j_{anode}$ = 0.16 mA cm$^{-2}$), 0.9 h (2.2 F mol$^{-1}$)
[b]Determined by $^1$H NMR analysis using 1,3,5-trimethoxybenzene as the internal standard
[c]Isolated yield

**7**), $^i$PrCO$_2$H (Table 1, entry 8) and BF$_3$•Et$_2$O (Table 1, entry 9) were all indispensable for achieving optimal reaction efficiency. Replacing $^i$PrCO$_2$H with AcOH (Table 1, entry 10) or CF$_3$CO$_2$H (Table 1, entry 11) led to a slight yield reduction. The yield of **5** dropped to 20% in the absence of $^i$PrCO$_2$H and BF$_3$•Et$_2$O (Table 1, entry 12). On the other hand, substituting HBF$_4$ (0.5 equiv) for both $^i$PrCO$_2$H and BF$_3$•Et$_2$O rescued the formation of **5** to a significant extent (Table 1, entry 13). We speculated that $^i$PrCO$_2$H and BF$_3$•Et$_2$O could complex to form a stronger protic acid, which is helpful for cathodic proton reduction and thus avoiding unwanted reduction of the substrates or products[48].

**Evaluation of substrate scope**. We next explored the scope of alkenes by using sulfamide **4** as the coupling partner (Table 2). The aryl ring in the 1,2-disubstituted alkene could be functionalized with an electronically diverse set of substituents, including Me (**9**), $^t$Bu (**10**), halogens (F, Cl, Br, I; **11–16**), and ester (**17** and **18**), at various positions of the phenyl ring. Alkenes carrying a 2,5-disubstituted phenyl ring were also found to be suitable substrates (**19** and **20**). The β-position of the styrenyl alkene showed broad tolerance for alkyl substituents of different sizes, such as Me (**21**), cyclohexyl (Cy; **22**), and $^t$Bu (**23**). Terminal alkenes were less-efficient substrates probably owing to the facile dimerization/oligomerization of these alkenes[44,49]. As an example, the reaction of 1,1-diphenylethylene with **4** afforded the desired product **24** in 40% yield. One the other hand, trisubstituted cyclic (**25**) and acyclic (**26** and **27**) alkenes reacted smoothly to afford the corresponding cyclic sulfamides in good to excellent diastereoselectivity. Meanwhile, 1,2-diamination of 1,3-dienes showed satisfactory regioselectivity in favor of the alkenyl moiety distal to the phenyl group (**28** and **29**). Note that triarylamine **6** was employed as redox catalyst to overcome the relatively low oxidation potentials of 1,3-dienes ($E_{p/2}$ = 1.29 V vs SCE) and avoid oxidizing the remaining alkene moiety in the products. Furthermore, the electrochemical alkene 1,2-diamination reaction was compatible with alkylbromide (**30** and **31**), alkylchloride (**32**), ester (**17**, **18**, **33**), sulfonic ester (**34**),

sulfonamide (**35**), amide (**36**), heterocycles such as furan (**37**) and thiophene (**38**), cyclic ether (**39**), and even oxidation-labile secondary and tertiary amines (**40–42**). Alkenes derived from estrone (**43**), fasudil (**44**), and quinine (**45**) reacted with similar success. The electron-rich amino groups in the cases of **40–42**, **44**, and **45** were masked as ammonium salts by the addition of HBF$_4$ to prevent oxidative decomposition.

The electrochemical method also proved capable of generating 1,2-diamine products that carry two differently decorated amino groups, or cyclic 1,2-diamines (Table 2). For example, we succeeded in the 1,2-diamination with a wide array of asymmetric sulfamides bearing a single alkyl group on one of its nitrogen atoms. In these cases, the alkyl substituent could be primary (**46**, **47**), secondary (**48**), tertiary (**49**, **50**), or functionalized with ester (**51**), CF$_3$ (**52**), alkylchloride (**53**), carboxylic acid (**54**), free alcohol (**55**), alkene (**56**), or alkyne (**57**). These asymmetric sulfamides reacted in a strictly regioselective manner. Notably, bridged bicyclic products (**58–60**) could be obtained by 1,2-diamination of cyclic sulfamide substrates. The structure of **59** was further confirmed by single crystal X-ray analysis.

**Gram scale synthesis and product transformations**. To further demonstrate the synthetic utility of our electrochemical method, we reacted alkene **61** or **62** with sulfamide **4**, **63**, or **64** on gram or even decagram scale and obtained the corresponding products (**21**, **22**, **48**, and **59**) in good yields (Fig. 3). Deprotection of theses cyclic sulfamides with HBr or hydrazine furnished diamines **65–67**, **69**, and **70**. Protection of the free amino group in **67** with Boc$_2$O resulted in the formation of **68**, whose two amino groups carries different substituents and therefore is amenable to further chemoselective derivatization. On the other hand, **48** could be converted to diamine **69**, also with differently decorated amino groups, through methylation and subsequent sulfonyl removal.

**Methods**
**Representative procedure for the synthesis of 5**. To a 10-mL three-necked round-bottomed flask were added sulfamide **4** (0.4 mmol, 2 equiv), triarylamine **1** (0.02 mmol, 0.1 equiv) and Et$_4$NPF$_6$ (0.2 mmol, 1 equiv). The flask was equipped

## Table 2 Substrate scope[a]

Reaction scheme: alkene (R$^1$, R$^2$, R$^3$) + sulfamide (R$^4$, R$^5$) → product with RVC | Pt electrodes, **1** (10 mol%), $^i$PrCO$_2$H (2 equiv), Et$_4$NPF$_6$ (1 equiv), BF$_3$•OEt$_2$ (0.5 equiv), MeCN/CH$_2$Cl$_2$ (1:2), RT, product, yield (dr >20:1 or specified)

- 52 examples
- Simple and stable materials
- Diastereoselective
- Differentiated N atoms
- Functional group tolerance

**Variation of alkene**

(±)-**9**, R = 4-Me, 86%
(±)-**10**, R = 4-$^t$Bu, 65%
(±)-**11**, R = 4-F, 76%
(±)-**12**, R = 4-Cl, 72%
(±)-**13**, R = 4-Br, 70%
(±)-**14**, R = 3-Br, 61%
(±)-**15**, R = 2-Br, 57%
(±)-**16**, R = 4-I, 61%
(±)-**17**, R = 4-CO$_2$Me, 45%
(±)-**18**, R = 3-CO$_2$Me, 52%
(±)-**19**, R = 2-F-5-Me, 82%
(±)-**20**, R = 2-Br-5-OMe, 61%

(±)-**21**, R = Me, 63%
(±)-**22**, R = Cy, 88%
(±)-**23**, R = $^t$Bu, 53%[b]

(±)-**24**, 40%

(±)-**25**, 77%

(±)-**26**, 37%[c]

(±)-**27**, 79% (9:1 dr)

(±)-**28**, 67%[d]

(±)-**29**, 47%[d]

(±)-**30**, 61%

(±)-**31**, 55%

(±)-**32**, 74% (8:1 d.r.)

(±)-**33**, 61%

(±)-**34**, 80%

(±)-**35**, 66%

(±)-**36**, 45%

(±)-**37**, 90%

(±)-**38**, 69%

(±)-**39**, 57%

(±)-**40**, 59%[e]

(±)-**41**, 95%[e]

(±)-**42**, 72%[e]

**43**, 76%
(1:1 dr relative to estrone)
*from estrone*

(±)-**44**, 82%[g]
*from fasudil*

**45**, 54%[g]
(1:0.8 dr relative to quinine)
*from quinine*

**Variation of sulfamide**[b]

(±)-**46**, R = $^n$Bu, 70%
(±)-**47**, R = Bn, 68%[f]
(±)-**48**, R = $^i$Pr, 88%
(±)-**49**, R = $^t$Bu, 67%

(±)-**50**, 91%

(±)-**51**, R = CO$_2$Et, 83%
(±)-**52**, R = CF$_3$, 70%
(±)-**53**, R = CH$_2$Cl, 70%

(±)-**54**, 47%

(±)-**55**, 54%

(±)-**56**, 79%

(±)-**57**, 79%

(±)-**58**, 44%

(±)-**59**, 77%

(X-ray)

(±)-**60**, 75%

[a]Reaction conditions: alkene (0.2 mmol), sulfamide (0.4 mmol), 0.9–3.7 h (2.0–8.7 F mol$^{-1}$). All yields are isolated yields
[b]Reaction with sulfamide (1.2 mmol) and BF$_3$•OEt$_2$ (0.2 mmol)
[c]Reaction without BF$_3$•OEt$_2$
[d]Reaction with **6** (10 mol %) as the catalyst
[e]Reaction with HBF$_4$ (0.3 mmol) instead of BF$_3$•OEt$_2$
[f]Reaction with sulfamide (0.8 mmol) and BF$_3$•OEt$_2$ (0.2 mmol)
[g]Reaction with HBF$_4$ (0.4 mmol) instead of BF$_3$•OEt$_2$. Cy, cyclohexyl; Ts, tosyl

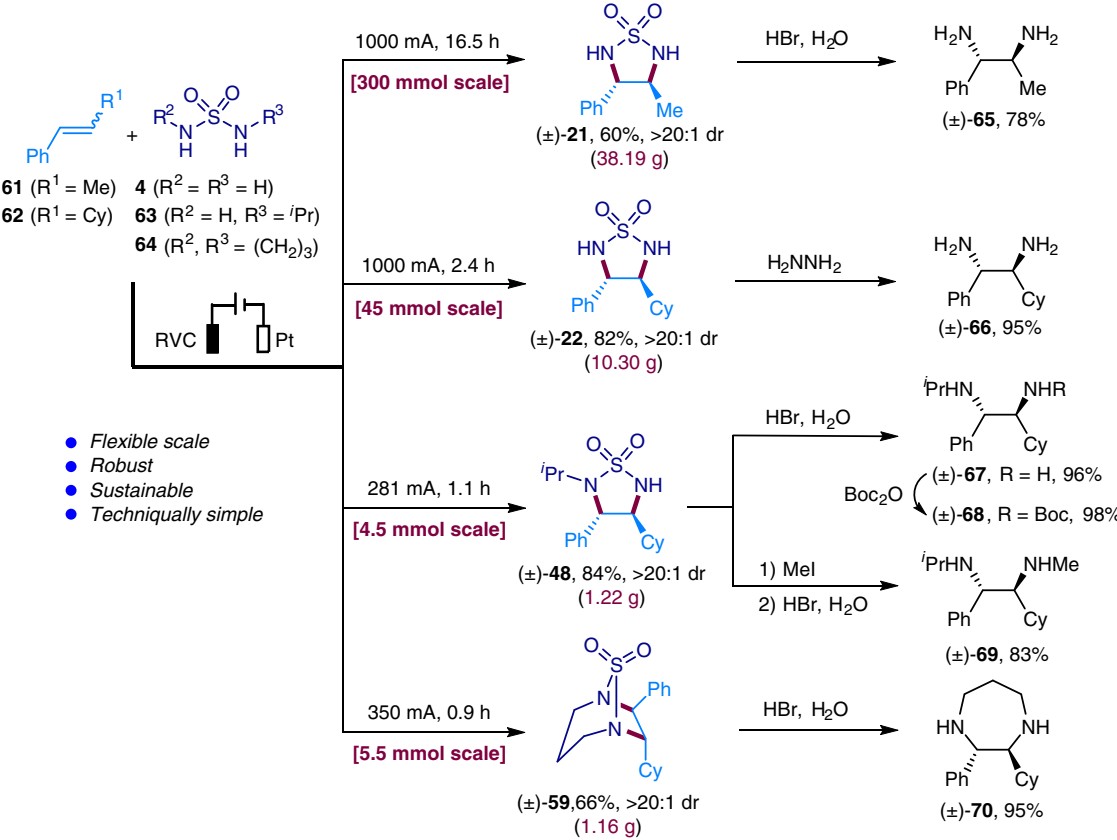

**Fig. 3** Gram scale synthesis and further product transformations. Gram scale synthesis of compounds **21**, **22**, **48**, and **59**, and their conversion to amines

with an RVC anode (100 PPI, 1 cm × 1 cm × 1.2 cm) and a platinum plate (1 cm × 1 cm) cathode. After flushing the flask with argon, MeCN (2 mL), $CH_2Cl_2$ (4 mL), alkene **2** (0.2 mmol, 1 equiv), $^{i}PrCO_2H$ (0.4 mmol, 2 equiv) and $BF_3 \cdot Et_2O$ (0.1 mmol, 0.5 equiv) were added sequentially. The constant current (12.5 mA) electrolysis was carried out at room temperature until complete consumption of **2** (monitored by TLC or $^1H$ NMR). Saturated $NaHCO_3$ solution was added. The resulting mixture was extracted with EtOAc (3 × 20 mL). The combined organic solution was dried over anhydrous $Na_2SO_4$ and concentrated under reduced pressure. The residue was separated by silica gel chromatography and the product **5** obtained in 72% yield by eluting with ethyl acetate/hexanes. All new compounds were fully characterized (See the Supplementary Methods).

## Data availability

The X-ray crystallographic coordinates for structures reported in this article have been deposited at the Cambridge Crystallographic Data Centre (CCDC), under deposition number CCDC 1938821 (**59**). The data can be obtained free of charge from The Cambridge Crystallographic Data Centre [http://www.ccdc.cam.ac.uk/data_request/cif]. The data supporting the findings of this study are available within the article and its Supplementary information files. Any further relevant data are available from the authors on request.

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

## Acknowledgements

We acknowledge the financial support of this research from MOST (2016YFA0204100), NSFC (No. 21672178), Program for Changjiang Scholars and Innovative Research Team in University and Fundamental Research Funds for the Central Universities.

## Author contributions

C.Y.C. and X.M.S. performed the experiments and analyzed the data. H.C.X. designed and directed the project and wrote the manuscript.

## Competing interests

The authors declare no competing interests.
