## [Peer Review File · Nature Communications]

Reviewers' comments:

Reviewer #1 (Remarks to the Author):

This manuscript reports a new electrochemical method for 1,2-diamination of internal olefins. The work is well-designed, -executed, and -reported. The results reported in this manuscript is very attractive for the following reasons.

(1) Stereoselectivity is major concern for diamination of olefins. The present reaction provides 1,2-diamine diastereoselectively.

(2) The 1,2-diamination reported in this manuscript is practical because it provides the desired products in gram scales without using external oxidant, transition metal catalyst, dangerous nitrogen source.

(3) The method provides useful 1,2 diamine compounds with broad functional group compatibility, which would contribute development of organic synthesis.

This reviewer recommends this manuscript to be published in Nature Communication after addressing the points indicated below.

(1) (page 2, Fig. 1a) General reaction schemes should be represented correctly. Some reactions were described as if scope of the reaction was limited to terminal olefin.

(2) (page 2, Fig 1c) R1 and R2 should be described in a limited expression (ex: R1 = Ar, vinyl R2 = alkyl) because the present reaction requires substituents which would affect the oxidation potentials of olefins.

(3) (Page 2, Fig 1) Abrams and Clayden reported cycloamination of prenyl carbamates and ureas promoted by Aryldiazonium Salts (Angew. Chem. Int. Ed. 2018, 13587). Davies and Lenori also reported photoredox imino functionalizations of olefins. These papers should be referred.

(4) (Page 2, Fig 1) Morofuji and Yoshida reported that diastereoselective introduction of two nitrogen substituents into an olefin via electrochemically generated radical cations of the olefins (J. Am. Chem. Soc. 2014, 136, 4496-4499, Table 3 entries 14, 15). The paper should be referred and add a statement about comparison of reaction modes. Diamination reaction reported by Morofuji and Yoshida consists of first intramolecular amination and second intermolecular amination. In contrast, the reaction reported in this manuscript consists of first intermolecular amination and second intramolecular amination.

(5) (Page 3, design plan) The authors have already reported dehydrogenative annulation of alkenes with diols for the synthesis of saturated O-heterocycles via electrochemically generated radical cation of olefins (ref. 34: Nat. Commun. 9, 3551 (2018).). Reaction mechanism and chemistry seem to be quite similar, and the present reaction is expansion of the dioxygenation described in the reported paper. Therefore, the authors should add independent statement of the reported paper.

(6) (page 7, table 2) The present diamination was not applied to terminal olefins such as 1,1-diphenylethene. The author should add a statement about the experimental results using terminal olefins.

(7) (page 8, L7) On the other hand, 31 ~
Is compound number 31 correct?

(8) (SI, p27) The relative stereochemistry of 27 should be determined using NOE experiment.

Reviewer #2 (Remarks to the Author):

Xu and co-workers described a electrocatalytic 1,2-diamination of 1,2-substituted alkenes for the synthesis of 1,2-diamines. Compared to the previous reported protocols, this scalable protocol could achieve good regio- and diastereo-selectivity under metal- and oxidant-free conditions. In consideration of the novelty and application prospect, I would recommend for the publication after addressing the following problems:

- 1) The Z/E ratio of the alkenes is not clear, the author should indicate clearly in the SI which isomer was used at the substrate'
- 2) A parallel experiment should be conducted with pure Z and E alkene as the substrate respectively to see whether the same diastereomer was obtained;
- 3) How the regio- and stereo-chemistry of products 45-56 were determined should be indicated in the SI;
- 4) Why the terminal alkene doesn't work under these conditions?
- 5) The explanation of entry 12 on page 5 is not clear. The sentences should be reorganized.
- 6) For the Gram scale synthesis, how long it take for the completion of different scale reactions should be indicated in the SI.
- 7) Did the authors try the asymmetric version of this reaction by using chiral reagents.

Reviewer #3 (Remarks to the Author):

Cai et al report an interesting electrocatalytic strategy for the 1,2-diamination of a large number of alkenes under ambient conditions using sulfamides as the amine source. Diamination of alkenes is an important organic reaction and this reviewer is glad to see novel electrochemical approaches to drive this reaction in a convenient manner. The authors screened many experimental parameters and investigated its versatility for many alkenes with varying substituents. Overall, this is a practical and useful electrosynthetic method for organic chemists, which seems interesting to the audience of Nature Communications. Nevertheless, this reviewer hopes the authors could address the following minor concerns before the consideration of acceptance.

1. Is there any special reason to use MeCN/CH₂Cl₂ as the solvent? Will a single solvent work?
2. The authors use the combination of iPrCOOH and BF₃Et₂O to prevent the potential reduction of substrates and products. However, this reviewer noticed a one-compartment cell was utilized for all the experiments. Can a divided two-compartment cell without iPrCOOH and BF₃Et₂O in working chamber produce similar results?
3. The authors used a current density of 0.16 mA/cm² for all the different experiments. What is the typical oxidation potential to achieve this current density? Will some of those substrates and/or products directly be oxidized at this potential?

Point-by-point response to the referees' comments is as follows:

Reviewer #1 (Remarks to the Author):

This manuscript reports a new electrochemical method for 1,2-diamination of internal olefins. The work is well-designed, -executed, and –reported. The results reported in this manuscript is very attractive for the following reasons.

(1) Stereoselectivity is major concern for diamination of olefins. The present reaction provides 1,2-diamine diastereoselectively.

(2) The 1,2-diamination reported in this manuscript is practical because it provides the desired products in gram scales without using external oxidant, transition metal catalyst, dangerous nitrogen source.

(3) The method provides useful 1,2 diamine compounds with broad functional group compatibility, which would contribute development of organic synthesis.

This reviewer recommends this manuscript to be published in Nature Communication after addressing the points indicated below.

Response: We are grateful to the reviewer for the recommendation.

(1) (page 2, Fig. 1a) General reaction schemes should be represented correctly. Some reactions were described as if scope of the reaction was limited to terminal olefin.

Response: Fig. 1a has been modified. The terminal alkenes in the third and fourth reaction have been replaced with internal alkenes.

(2) (page 2, Fig 1c) R1 and R2 should be described in a limited expression (ex: R1 = Ar, vinyl R2 = alkyl) because the present reaction requires substituents which would affect the oxidation potentials of olefins.

Response: Fig 1c has been revised according to the reviewer suggestion.

(3) (Page 2, Fig 1) Abrams and Clayden reported cycloamination of prenyl carbamates and ureas promoted by Aryldiazonium Salts (Angew. Chem. Int. Ed. 2018, 13587). Davies and Lenori also reported photoredox imino functionalizations of olefins. These papers should be referred.

Response: Clayden and Lenori papers have been added as references 17 and 18.

(4) (Page 2, Fig 1) Morofuji and Yoshida reported that diastereoselective introduction of two nitrogen substituents into an olefin via electrochemically generated radical cations of the olefins (J. Am. Chem. Soc. 2014, 136, 4496-4499, Table 3 entries 14, 15). The paper should be referred and add a statement about comparison of reaction modes. Diamination reaction reported by Morofuji and Yoshida consists of first intramolecular amination and second intermolecular amination. In contrast, the reaction reported in this manuscript consists of first intermolecular amination and second intramolecular amination.

Response: Yoshida's paper has been referenced as reference 35. The following statement has been added to the manuscript text: Yoshida reported isolated examples of alkene diamination through intramolecular trapping of electrochemically generated alkene radical cations.

(5) (Page 3, design plan) The authors have already reported dehydrogenative annulation of alkenes with diols for the synthesis of saturated O-heterocycles via electrochemically generated radical cation of olefins (ref. 34: Nat. Commun. 9, 3551 (2018).). Reaction mechanism and chemistry seem to be quite similar, and the present reaction is expansion of the dioxygenation described in the reported paper. Therefore, the authors should add independent statement of the reported paper.

Response: The following statement has been added to the manuscript text: Inspired by our previously work on electrochemical alkene dioxygenation reactions.

(6) (page 7, table 2) The present diamination was not applied to terminal olefins such as 1,1-diphenylethene. The author should add a statement about the experimental results using terminal olefins.

Response: 1,1-Diphenylethylene has been added as an example (product **24**, Table 2). The following statement has been added to the manuscript text: Terminal alkenes were less efficient substrates probably due to the facile dimerization/oligomerization of these alkenes. As an example, the reaction of 1,1-diphenylethylene with **4** afforded the desired product **24** in 40% yield.

(7) (page8, L7) On the other hand, 31 ~
Is compound number 31 correct?

Response: This compound is correct.

(8) (SI, p27) The relative stereochemistry of 27 should be determined using NOE

experiment.

Response: NOE of compound **27** has been included in the SI.

Reviewer #2 (Remarks to the Author):

Xu and co-workers described a electrocatalytic 1,2-diamination of 1,2-substituted alkenes for the synthesis of 1,2-diamines. Compared to the previous reported protocols, this scalable protocol could achieve good regio- and diastereo-selectivity under metal- and oxidant-free conditions. In consideration of the novelty and application prospect, I would recommend for the publication after addressing the following problems:

Response: We are grateful to the reviewer for the recommendation.

1) The Z/E ratio of the alkenes is not clear, the author should indicate clearly in the SI which isomer was used at the substrate'

Response: The stereochemical information for the alkenes have been added to the SI.

2) A parallel experiment should be conducted with pure Z and E alkene as the substrate respectively to see whether the same diastereomer was obtained;

Response: The suggested experiments have been carried out with alkene **2**. The following statement has been added to the manuscript text: Independent reaction using a pure *E*- or *Z*-isomer of **2** afforded the same *trans* diastereomer **5** in 65% and 69% yield, respectively.

3) How the regio- and stereo-chemistry of products 45-56 were determined should be indicated in the SI;

Response: The regiochemistry were determined by NMR analysis. The NH proton is coupled with the alpha C-H proton. The stereochemistry is assumed to be the same as that of compound **5**.

4) Why the terminal alkene doesn't work under these conditions?

Response: Terminal alkenes are less efficient in radical cation reactions probably due to the facile electron-transfer induced dimerization/oligomerization. Similar trend has been observed under photochemical conditions (*Nat. Chem.* **2014**, *6*, 720).

1,1-Diphenylethylene has been added as an example (product **24**, Table 2). The following statement has been added to the manuscript text: Terminal alkenes were less efficient substrates probably due to the facile dimerization/oligomerization of these alkenes. As an example, the reaction of 1,1-diphenylethylene with **4** afforded the desired product **24** in 40% yield.

5) The explanation of entry 12 on page 5 is not clear. The sentences should be reorganized.

Response: The sentence has been reworded.

6) For the Gram scale synthesis, how long it take for the completion of different scale reactions should be indicated in the SI.

Response: The reaction time has been added to the SI.

7) Did the authors try the asymmetric version of this reaction by using chiral reagents.

Response: Research on the asymmetric version of this diamination reaction is ongoing in our laboratory.

Reviewer #3 (Remarks to the Author):

Cai et al report an interesting electrocatalytic strategy for the 1,2-diamination of a large number of alkenes under ambient conditions using sulfamides as the amine source. Diamination of alkenes is an important organic reaction and this reviewer is glad to see novel electrochemical approaches to drive this reaction in a convenient manner. The authors screened many experimental parameters and investigated its versatility for many alkenes with varying substituents. Overall, this is a practical and useful electrochemical method for organic chemists, which seems interesting to the audience of Nature Communications. Nevertheless, this reviewer hopes the authors could address the following minor concerns before the consideration of acceptance.

Response: We are grateful to the reviewer for the recommendation.

1. Is there any special reason to use MeCN/CH₂Cl₂ as the solvent? Will a single solvent work?

Response: The use of mixed solvent led to optimal yields. The reaction of **2** with **4** in MeCN afforded the desired **5** in only 50% yield. These results have been included in Table 1, entry 3.

2. The authors use the combination of *i*PrCOOH and BF₃Et₂O to prevent the potential reduction of substrates and products. However, this reviewer noticed a one-compartment cell was utilized for all the experiments. Can a divided two-compartment cell without *i*PrCOOH and BF₃Et₂O in working chamber produce similar results?

Response: Undivided cells are usually more desirable than divided cells for organic electrosynthesis because of the simplicity of one-compartment cells. Electrolysis in divided cell is often associated with high resistance and more difficult for reaction scale up. We have carried out the reaction of **2** with **4** using a divided cell with a glass frit separator. Specifically, the anode chamber was charged with compounds **2** and **4**, solvent (MeCN/CH₂Cl₂), catalyst **1**, and Et₄NPF₆. The cathodic compartment was charged with solvent (MeCN/CH₂Cl₂) and Et₄NPF₆ with or without *i*PrCO₂H (2 equiv)/BF₃•Et₂O (0.5 equiv). With the acid additives in the cathodic chamber, the initial cell potential was 56 V and increased to 300 V during the later stage of the reaction. This reaction afforded compound **5** in 55% yield. Without acidic additives in the cathode chamber, the initial potential was 300 V and increased to 1000 V (the maximum for the instrument). In this case, the yield of **5** was 28%. In contrast, the cell potential was around 6 V when an undivided cell was used.

3. The authors used a current density of 0.16 mA/cm² for all the different experiments. What is the typical oxidation potential to achieve this current density? Will some of those substrates and/or products directly be oxidized at this potential?

Response: The reaction employed a redox catalyst. The anode potential should be close to that of the catalyst employed. The potential of the catalyst is lower than those of the substrates and products. As a result, the substrates and products are not oxidized on the anode. The direct oxidation of the substrate on the electrode is inefficient in promoting the diamination reaction as demonstrated in Table 1 (entry 7).

REVIEWERS' COMMENTS:

Reviewer #1 (Remarks to the Author):

This manuscript is a revised version reporting electrochemical diamination of olefins. The authors have addressed all the points of the reviewers' comments. This manuscript is now suitable to be published in Nature Communications.

Reviewer #2 (Remarks to the Author):

After careful review of the revised manuscript by Xu and Co-workers, I believe all the concerns from the reviewers have been addressed properly. Hence, I recommend it for the publication in Nature Communications

Point-by-point response to the referees' comments is as follows:

REVIEWERS' COMMENTS:

Reviewer #1 (Remarks to the Author):

This manuscript is a revised version reporting electrochemical diamination of olefins. The authors have addressed all the points of the reviewers' comments. This manuscript is now suitable to be published in Nature Communications.

Reviewer #2 (Remarks to the Author):

After careful review of the revised manuscript by Xu and Co-workers, I believe all the concerns from the reviewers have been addressed properly. Hence, I recommend it for the publication in Nature Communications.

Response: No issues have been raised by the referees.